# Experience of Sexual Orientation Microaggression among Young Adult Lesbian, Gay, and Bisexual Individuals in Taiwan: Its Related Factors and Association with Mental Health Problems

**DOI:** 10.3390/ijerph182211744

**Published:** 2021-11-09

**Authors:** Ching-Shu Tsai, Yu-Te Huang, Cheng-Fang Yen

**Affiliations:** 1School of Medicine, Chang Gung University, Taoyuan 33302, Taiwan; jingshu@cgmh.org.tw; 2Department of Child and Adolescent Psychiatry, Chang Gung Memorial Hospital, Kaohsiung Medical Center, Kaohsiung 83301, Taiwan; 3Department of Social Work and Social Administration, The University of Hong Kong, Hong Kong RM543, China; Yuhuang@hku.hk; 4Department of Psychiatry, School of Medicine College of Medicine, Kaohsiung Medical University, Kaohsiung 80752, Taiwan; 5Department of Psychiatry, Kaohsiung Medical University Hospital, Kaohsiung 80756, Taiwan; 6College of Professional Studies, National Pingtung University of Science and Technology, Pingtung 91201, Taiwan

**Keywords:** microaggression, psychological well-being, sexual orientation, family support

## Abstract

Experiences of sexual orientation microaggression (SOM) are prevalent in lesbian, gay, and bisexual (LGB) individuals. The aims of this quantitative cross-sectional survey study were to examine the factors, including demographics, sexual orientation characteristics, and perceived social support related to SOM, as well as the relationships of SOM with anxiety, depression, and suicidality among young adult LGB individuals in Taiwan. In total, 1000 self-identified young adult LGB individuals (500 men and 500 women) participated in this study. The experience of SOM was assessed using the Sexual Orientation Microaggression Inventory. We also collected demographic and sexual orientation characteristics; perceived general family support, using the Family APGAR Index; anxiety on the State-Trait Anxiety Inventory; depression on the Center for Epidemiological Studies-Depression Scale; and suicidality on the suicidality module of the Kiddie-SADS. The factors related to SOM and the associations of SOM with anxiety, depression, and suicidality were examined using multivariate linear regression analysis. The results indicated that males experienced greater SOM than females, and that younger age of identification of sexual orientation and perceived lower general family support were significantly associated with greater SOM. Greater SOM was significantly associated with greater anxiety, depression, and suicidality. The experiences of SOM in LGB individuals with mental health problems warrant assessment and intervention that take the related factors into account.

## 1. Introduction

Lesbian, gay, and bisexual (LGB) individuals may experience multiple forms of prejudice and discrimination based on sexual orientation [1,2]. Compared with overt stigmatizing attitudes and behaviors such as bullying, sexual orientation microaggression (SOM) is a more subtle and covert prejudice against LGB individuals [3,4]. According to Sue et al. [5], microaggression is defined as “brief and commonplace daily verbal, behavioral, or environmental indignities, whether intentional or unintentional, that communicate hostile, derogatory, or negative slights or insults” (p. 72). The concept of microaggression has been applied from racism to the stigmatizing experiences in marginalized groups, including LGB individuals [3,4,6]. According to Swann et al. [4], three forms of SOM rooted in heterosexism are identified: microassault, microinsult, and microinvalidation [3,4,6]. Microassaults refer to discriminatory verbal or non-verbal behaviors against LGB individuals; microinsults refer to subtle slights due to sexual minority identity; and microinvalidations refer to the engagement in communications that nullify the stigmatized experiences of a LGB individual [3,4,6].

SOM may result in negative impacts on LGB individuals. Research has demonstrated that SOM is significantly associated with depression [3], anxiety [3,7], smoking cigarettes [8], negative feelings toward sexual identity [9], low self-esteem [7,9], and non-response to psychotherapy among LGB individuals [10]. Young adulthood is a phase of the life span from adolescence to full-fledged adulthood where the individuals become more independent and explore various life possibilities [11]. Examining SOM and its relationship with mental health problems among young adult LGB individuals is of uttermost importance.

### 1.1. Rationale for Given Study

There are several issues regarding SOM warranting further study. First, SOM has not been examined among LGB individuals in non-Western societies. Asian societies have a lower tolerance for sexual minority compared with Western societies [12]. Overt and covert aggressions toward LGB individuals in Asian-Pacific regions are serious health issues [13]. Although significant associations between SOM and poor mental health have been found among LGB individuals in Western societies [3,7,8], further study is needed on the associations among LGB individuals in non-Western societies. Second, identifying the individual and environmental factors related to SOM in LGB individuals can provide empirical evidence for developing intervention programs. Although the results of previous studies examining the gender difference in social stigmas in sexual minorities were mixed [14], gay and bisexual men usually reported a higher risk of sexuality-related bullying victimization than lesbian and bisexual women [15]. Regarding sexual orientation characteristics, research found that early identification of sexual orientation was significantly associated with sexuality-related bullying victimization [16]; bisexual individuals were more likely to experience sexuality-related stigma compared with other sexual minorities [17]. In addition to the individual factors, family is an important microsystem that may protect and hurt an individual [18]. Research has demonstrated that a high proportion of LGB individuals experienced family-level interpersonal and environmental microaggressions [19], whereas high general family support can protect LGB individuals from sexuality-related bullying victimization [16]. However, no study examined the relationships of various genders, sexual orientation, age of identification of sexual orientation, and perceived general family support with SOM in LGB individuals.

### 1.2. Aims

The aims of this cross-sectional survey study were to examine the factors, including demographics, sexual orientation characteristics, and perceived social support related to SOM, and the relationships of SOM with anxiety, depression, and suicidality among young adult LGB individuals in Taiwan. We hypothesized that male sexual minorities report greater SOM than females; bisexuals report greater SOM than gay men and lesbians; earlier identification of sexual orientation is associated with greater SOM; and perceived lower general family support is associated with greater SOM. We also hypothesized that after controlling for the effects of demographics, greater SOM is associated with greater anxiety, depression, and suicidality.

## 2. Materials and Methods

### 2.1. Participants and Procedure

The participant inclusion criteria were individuals who identified their sexual orientation as being gay/lesbian or bisexual, aged between 20 and 30 years, and living in Taiwan. Participants were recruited by posting an online advertisement on social media, including *Facebook*, *Twitter*, and *LINE* (a direct messaging app); the Bulletin Board System; and the home pages of three health promotion and counseling centers for LGB individuals, from August 2018 to July 2020. Anyone who intended to participate in the study telephoned the research assistants. The research assistant ensured the eligibility of potential participants against recruitment criteria, explained the study aims and procedures to them, and scheduled the time for completing the study questionnaires individually in the study room. The research assistants evaluated the participants in the on-site study room to determine whether they had impaired intellect or showed signs of alcohol and substance use that might interfere with understanding the study’s purpose or completing the questionnaire. In total, 1000 participants (500 males and 500 females) participated in the study. No participants were excluded. Informed consent was obtained from all participants prior to the assessment. Because the present study assessed participants’ mental health and suicidality, we provided participants with information about mental health resources if needed. The study was approved by the Institutional Review Board of Kaohsiung Medical University Hospital (KMUHIRB-F(II)-20180018).

### 2.2. Measures

#### 2.2.1. Sexual Orientation Microaggression Inventory (SOMI)

The traditional Chinese version [20] of the SOMI [4] contains 19 items, assessing the experiences of SOM in the last six months among LGB individuals with four trait factors, including anti-LGB attitudes and expressions, denial of homosexuality, heterosexualism, and societal disapproval [4]. The SOMI structure generally follows the types of microaggressions (microassaults, microinvalidations, microinsults) [4]. The SOMI items are rated on a five-point Likert-type scale (score 1 = not at all; score 5 = about every day); therefore, a higher SOMI score indicates a higher level of microaggression. The SOMI was found to have a bifactor structure in its psychometric evidence; there is a general factor in the SOMI apart from the four trait factors mentioned above [4]. The SOMI was translated into the traditional Chinese version for Taiwanese LGB individuals, using the standard forward-, backward-, and pretest-step methods [21]. The traditional Chinese version of the SOMI had a bifactor structure the same as the original instrument, and acceptable internal consistency (Cronbach’s α = 0.90) and concurrent validity (correlations with familial stigma and psychological inflexibility: r = 0.336 and 0.262, respectively; *p* < 0.001) [20].

#### 2.2.2. Demographic and Sexual Orientation Factors

We collected the participants’ gender, age, education level (high school or below vs. college or above), sexual orientation (gay/lesbian or bisexual), and age of identification of sexual orientation (“When did you firstly identify yourself as a gay/lesbian or bisexual?”).

#### 2.2.3. Chinese Version of the Family APGAR Index

The 5-item Chinese version [22] of the Family APGAR Index [23] was applied to measure participants’ perceived general family support for the components of adaptability, partnership, growth, affection, and resolve, in the most recent month. A higher total score on the APGAR represented a higher level of perceived general family support. The Chinese version of the Family APGAR Index had acceptable discriminatory validity for social adaptability [22] and congruent validity with a significant correlation with general health state [24]. Cronbach’s α in the present study was 0.86.

#### 2.2.4. State Subscale on the Chinese Version of the State-Trait Anxiety Inventory (STAI-S)

We used the 20 items from the self-administered Chinese version of the STAI-S to assess participants’ current anxiety symptoms [25,26]. The items were graded on a 4-point Likert scale, with scores ranging from 1 (not at all) to 4 (very much so). Higher total STAI-S scores indicated more severe anxiety. The Chinese version of the STAI-S had acceptable test–retest reliability (Pearson’s r = 0.76), internal reliability (Cronbach’s α = 0.91), criterion validity (correlation with the Hamilton Anxiety Rating Scale: r = 0.69), and construct validity [27]. Cronbach’s alpha for the STAI-S in the present study was 0.89.

#### 2.2.5. Mandarin Chinese Version of the Center for Epidemiological Studies-Depression Scale (MC-CES-D)

We used the 20-item self-administered MC-CES-D to assess the frequency of depressive symptoms in the month preceding the study [28,29]. The items were graded on a 4-point scale. Higher total MC-CES-D scores indicated more severe depression. The MC-CES-D had good internal consistency (Cronbach’s α = 0.90), 1-week test-retest reliability (intraclass correlation reliability = 0.93), congruent validity (area under the receiver operative characteristic curves for major depressive disorder = 0.88–0.90) [30], and construct validity [31]. Cronbach’s alpha for the MC-CES-D in the present study was 0.93.

#### 2.2.6. Suicidality Module of the Epidemiological Version of the Kiddie-Schedule for Affective Disorders and Schizophrenia (Kiddie-SADS)

A 5-item questionnaire from the epidemiological version of the Kiddie-SADS [32] was used to assess the frequency of suicide ideation and attempts in the preceding month [33]. Each question elicited a “yes” or “no” response. The suicidality module had acceptable congruent validities (correlations with depression, anxiety, and hostility: r = 0.45–0.67) [34]. Cronbach’s alpha for the suicidality module of the Kiddie-SADS in the present study was 0.81.

### 2.3. Statistical Analysis

With the use of descriptive statistics, including mean (SD) and frequency (percentage), the participants’ demographics, sexual orientation factors, perceived general family support, SOM, anxiety, depression, and suicidality were analyzed. The absolute values of skewness and kurtosis of continuous variables were 0.071–1.310 and 0.122–1.139, respectively; according to Kim [35], the continuous variables in this study were normally distributed. The associations of demographics, sexual orientation factors, and perceived general family support with SOM were examined using multivariate linear regression analysis. The associations of SOM with anxiety, depression, and suicidality were also examined using multivariate linear regression analysis, controlling for the effects of demographics. A value of *p* < 0.05 was considered statistically significant. The data analyses were done using the IBM SPSS 20.0 (IBM Corp., Armonk, NY, USA).

## 3. Results

All 1000 participants completed the research questionnaire without omission. Their demographics, sexual orientation factors, perceived general family support, SOM, anxiety, depression, and suicidality are presented in Table 1. The mean (SD) age was 24.6 years (3.0 years); 89.1% had an educational degree of college or above; 57% identified themselves as gays/lesbians; the mean (SD) age to firstly identify sexual orientation was 14.5 (3.9) years old. The mean score (SD) of SOM was 42.0 (11.6). The severities of anxiety, depression, and suicidality were 40.8 (12.7), 18.8 (11.2), and 1.1 (1.5), respectively.

Table 2 shows the results of multivariate linear regression analysis examining the associations of demographics, sexual orientation factors, and perceived general family support with SOM. The result indicated that males had greater SOM than females; younger age of identification of sexual orientation was significantly associated with greater SOM; and perceived lower general family support was significantly associated with greater SOM. The condition index was 29.981, indicating no problem of collinearity.

Table 3 shows the results of multivariate linear regression analysis examining the associations of SOM with anxiety, depression, and suicidality. The result indicated that after controlling for the effects of demographics, greater SOM was significantly associated with greater anxiety, depression, and suicidality. The condition index was 25.230, indicating no problem of collinearity.

## 4. Discussion

The present study found that gender, age of identification of sexual orientation, and perceived general family support significantly related to the level of SOM in LGB individuals. Greater SOM was significantly associated with greater anxiety, depression, and suicidality in LGB individuals.

The present study demonstrated that gay and bisexual men reported greater SOM than lesbian and bisexual women. The experiences of SOM reflect the level of endorsement for heterosexism in this society; therefore, the stereotypes and expectations for a gender in society may influence the gender difference in the expression and context of SOM [36]. People in Taiwan have been deeply influenced by Confucianism, which considers that men should get married and have children to maintain family bloodlines, and men who are unmarried and have no offspring will be seen as a failure in observing filial piety [37,38]. Homosexuality is regarded as a challenge to the family obligations mandated in Confucianism [39]. Therefore, gay and bisexual men may experience greater SOM than lesbian and bisexual women. A previous qualitative study found that compared with lesbians, bisexual-identified women reported several bisexual-specific SOM, including hostility; denial/dismissal; unintelligibility; pressure to change; lesbian, gay, bisexual and transgender legitimacy; dating exclusion; and hypersexuality [40]. Further studies are needed to deeply explore the gender differences in the specific contexts of SOM experienced by LGB individuals.

We found that younger age of identification of sexual orientation was significantly associated with greater SOM in LGB individuals. First identifying as a sexual minority is one of the major developmental milestones for sexual minorities [41,42]. Early timing of sexual orientation development may increase the risk for LGB individuals to face stigma related to sexual minority orientation and experience negative mental health outcomes such as depression and anxiety [43,44]. Moreover, individuals reaching sexual orientation developmental milestones earlier might have less access to supportive resources [44]; LGB individuals who lack positive social support may encounter difficulties in communicating with the enactors regarding the bias in SOM.

This study demonstrated that perceived lower general family support was significantly associated with greater SOM in LGB individuals. Research found that a high proportion of LGB individuals experienced SOM enacted by family members [19]. Moreover, family-level SOM increased the risk of polyvictimization (including property victimization, bias and non-bias-motivated forms of physical assault, child maltreatment, sexual victimization, intimate partner violence, school-based bullying, cyberbullying, and indirect or witnessed forms of victimization) in LGB individuals [45]. The results of previous studies indicated that unfriendly family environments may not only be the source of SOM but also increase the risk for the LGB individuals to encounter SOM outside family environments. Young LGB individuals may need family assistance to meet developmental demands and to guide their personal experiences in various domains (e.g., interpersonal, romantic) and settings (e.g., school, work). LGB individuals with inadequate support from families may spend much time with and seek assistance from the individuals outside family environments; however, the chances of experiencing SOM may also increase simultaneously. Alternatively, LGB individuals experiencing greater SOM may need more family support for coping with such challenges; the chances of reporting perceived low family support may also increase.

Congruent with the results of previous studies [3,7], the present study demonstrated the significant associations of SOM with greater anxiety, depression, and suicidality in LGB individuals. Although SOM is not the single reason accounting for mental health problems, SOM might negatively impact LGB individuals in several ways. For example, the enactors of SOM might view their own behavior as harmless, unremarkable, or well intentioned; targets of SOM often face difficulties in communicating with the enactors what they feel about the hostility, derogation, and insults in SOM [46]. The dilemma might confuse or even demoralize LGB individuals and compromise their emotional regulation. The cross-sectional design of this study could not rule out the possibility that mental health problems might increase LGB individuals’ sensitivity to the existence of SOM.

This study is one of the first to examine the factors related to SOM and the relationship between SOM and mental health problems among LGB individuals in Asian societies. Because of the significant relationships between SOM and mental health problems in LGB individuals, individual-level and environment-level interventions to reduce SOM and related psychological harms are needed. Regarding the individual-level interventions, the experiences of SOM should be routinely surveyed among LGB individuals with mental health problems, helping the targets of SOM to develop alternative cognitive and emotional coping strategies for the experiences of SOM. The gender and sexual factors related to SOM identified in this study warrant being integrated into intervention programs. Regarding the environmental-level interventions, governments and health professionals should develop intervention programs to reduce SOM in the public [13,47]. Enhancing general family support may help to reduce SOM in LGB individuals.

### Limitations of the Given Study

There are some limitations in the present study. First, the cross-sectional study design limited the temporal relationships among SOM, general family support, and mental health problems. Further prospective studies are needed to examine the temporal relationships among the variables. Second, the present sample comprised young adult LGB individuals. Therefore, it is unclear whether the results of this study could be generalized to populations of other age ranges. Moreover, research found that social context (e.g., family cultural practices, community, etc.) can shape how a person experiences microaggressions from others [48]; whether the results of this study can be generalized to non-western contexts warrants further study. Third, all the data collected in the present study were self-reported. Therefore, single-rater biases cannot be fully controlled for. Fourth, this study inquired about participants’ gender identities by the binary of man and woman but did not include the options of transgender, gender nonbinary, or genderqueer. Research has found that sexual and gender minority identities have intersectional impacts on health [49], behaviors [50], and risk of intimate partner violence [51]. Fifth, the present study examined the association of SOM with perceived general family support but not family acceptance and rejection of sexual orientation. Research found that family acceptance and rejection of sexual orientation is crucial to the health and well-being of LGB individuals, especially LGB youth [52]. The role of family acceptance and rejection of sexual orientation in the experience of SOM among LGB individuals warrants further study.

## 5. Conclusions

Gender, age of identification of sexual orientation, and perceived general family support related to the level of SOM experienced by young adult LGB individuals. The experiences of SOM were significantly associated with anxiety, depression, and suicidality. Mental health professionals should routinely assess the experience of SOM among LGB individuals who suffer from mental health problems and help them develop effective strategies to cope with SOM. Individual and environmental factors that related to SOM warrant consideration in developing intervention programs.

## Figures and Tables

**Table 1 ijerph-18-11744-t001:** Participants’ characteristics (N = 1000).

Variables	*n* (%)	Mean (SD)	Range
Gender			
Female	500 (50)		
Male	500 (50)		
Age (years)		24.6 (3.0)	20–30
Education level			
High school or below	109 (10.9)		
College or above	891 (89.1)		
Sexual orientation			
Bisexual	430 (43)		
Gay/lesbian	570 (57)		
Age of identification of sexual orientation (years)		14.5 (3.9)	5–29
Perceived general family support		136 (3.6)	5–20
Sexual orientation microaggression		42.0 (11.6)	19–79
Anxiety		40.8 (12.7)	20–79
Depression		18.8 (11.2)	0–57
Suicidality		1.1 (1.5)	0–5

**Table 2 ijerph-18-11744-t002:** Factors related to sexual orientation microaggression: Multivariate linear regression analysis.

Variables	B (SE)	*p*
Male ^a^	2.444 (0.765)	0.001
Age	0.058 (0.125)	0.645
Education degree of college or above ^b^	0.242 (1.180)	0.838
Gays/lesbians ^c^	−1.397 (0.845)	0.098
Age of identification of sexual orientation	−0.259 (0.102)	0.011
Perceived general family support	−0.385 (0.101)	<0.001

^a^: female as reference; ^b^: education degree of high school or below as reference; ^c^: bisexuality as reference.

**Table 3 ijerph-18-11744-t003:** Association of sexual orientation microaggression with anxiety, depression, and suicidality: Multivariate linear regression analysis.

Variables	Anxiety		Depression		Suicidality	
	B (SE)	*p*	B (SE)	*p*	B (SE)	*p*
Male ^a^	−2.069 (0.778)	0.008	−1.498 (0.681)	0.028	−0.190 (0.095)	0.044
Age	0.104 (0.131)	0.428	−0.057 (0.115)	0.622	−0.016 (0.016)	0.301
Education degree of college or above ^b^	−0.892 (1.257)	0.478	−1.814 (1.099)	0.099	−0.180 (0.153)	0.237
Sexual orientation microaggression	0.302 (0.034)	<0.001	0.306 (0.029)	<0.001	0.027 (0.004)	<0.001

^a^: female as reference; ^b^: education degree of high school or below as reference.

## Data Availability

The data will be available upon reasonable request to the corresponding authors.

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
