# Peer review of "Experience of Sexual Orientation Microaggression among Young Adult Lesbian, Gay, and Bisexual Individuals in Taiwan: Its Related Factors and Association with Mental Health Problems"

_ijerph, 2021, doi:10.3390/ijerph182211744_

Round 1

Reviewer 1 Report

Dear authors

Many thanks for submitting this manuscript. I thoroughly enjoyed reading the text. Please find attached my comments which will hopefully strenghten the paper. 

Title

The title does not fully relate to the entire population under investigation. You need to add something to the title recognising the bisexual community in which you are also examining. 

Abstract

In line 18, please mention that this is a quantitative article so the text reads "this quantitative cross-sectional study

In line 30, please include "and intervention which take the related factors into account" 

Please ensure you include all quantitative scales and measures in the abstract. 

Introduction

In line 46 you state "subtle snubs." What does this mean ???

Before line 57, please put the title: Rationale for Given Study

Before line 79, please put title: Aims

Material and Methods

In line 97, remove for and replace with against so that it reads: "participants against recruitment criteria" 

In line 156, I assume it should be Kiddie-SADS and not MC-CES-D, Please confirm. 

Discussion

In line 206, please remove s between regarded and as. 

In terms of limitations, there should be something about how it is not generalisable outside of the non-western context. Please include a sentence on this here. I would also like to see the sub-heading : Limitations of the Given Study added before line 250. 

Overall, this is a really enjoyable and interesting study to read and I feel that these additions will make this study even easier and enjoyable for the readership of the journal to read. 

Author Response

We appreciated your valuable comments. As discussed below, we have revised our manuscript with underlines based on your suggestions. Please let us know if we need to provide anything else regarding this revision.

Comment 1

Title

The title does not fully relate to the entire population under investigation. You need to add something to the title recognising the bisexual community in which you are also examining. 

Response

Thank you for your reminding. We revised the title into “Experience of Sexual Orientation Microaggression Among Young Adult Lesbian, Gay, and Bisexual Individuals in Taiwan: Its Related Factors and Association with Mental Health Problems.” Please refer to line 2-5.

Comment 2

Abstract

In line 18, please mention that this is a quantitative article so the text reads "this quantitative cross-sectional study

Response

We added “quantitative” into line 19 of the revised manuscript.

Comment 3

In line 30, please include "and intervention which take the related factors into account" 

Response

We included “and intervention which take the related factors into account” into line 33.

Comment 4

Please ensure you include all quantitative scales and measures in the abstract. 

Response

We added the Family APGAR Index, State-Trait Anxiety Inventory, Center for Epidemiological Studies-Depression, and suicidality module of the Kiddie-SADS into Abstract in the revised manuscript. Please refer to line 25-27.

Comment 5

Introduction

In line 46 you state "subtle snubs." What does this mean ???

Response

We replaced “snubs” by “slights” in the revised manuscript to make it easily understood. Please refer to line 49.

Comment 6

Before line 57, please put the title: Rationale for Given Study

Response

We added it into Introduction of the revised manuscript. Please refer to line 60.

Comment 7

Before line 79, please put title: Aims

Response

We added it into Introduction of the revised manuscript. Please refer to line 83.

Comment 8

Material and Methods

In line 97, remove for and replace with against so that it reads: "participants against recruitment criteria" 

Response

We removed “for” and replaced with “against” in the revised manuscript. Please refer to line 102.

Comment 9

In line 156, I assume it should be Kiddie-SADS and not MC-CES-D, Please confirm. 

Response

Thank you for your reminding. We corrected the error. Please refer to line 164.

Comment 10

Discussion

In line 206, please remove s between regarded and as. 

Response

Thank you for your reminding. We corrected the error. Please refer to line 216.

Comment 11

In terms of limitations, there should be something about how it is not generalisable outside of the non-western context. Please include a sentence on this here.

Response

We added the sentence below into Limitations. Please refer to line 278-281.

“Moreover, research found that social context (e.g., family cultural practices, community, etc.) can shape how a person experiences microaggressions from others [48]; whether the results of this study can be generalized to the non-western contexts warrants further study.”

Comment 12

I would also like to see the sub-heading: Limitations of the Given Study added before line 250. 

Response

We added it into the revised manuscript. Please refer to line 272.

Reviewer 2 Report

This paper presented a study on the relationship between sexual orientation microaggressions and mental health in a sample from Taiwan. It tested these relationships, which are somewhat established in the American context, with a sample that had not been studied before. The authors also tested predictors of experiences of microaggressions to show what kinds of characteristics might be associated with higher frequency of these experiences.

In general, the study was very well performed and executed. The use of several existing instruments, the collection of a large sample, and the use of a multitude of recruitment procedures added to the study's quality. The analyses were appropriate to the data collected and offered important results. The authors appeared to cover the literature on SOM, particularly other research that had connected experiences of SOM with other outcomes.

My biggest concern is in the measurement of family support. I want to ask the authors to exercise caution in how they draw conclusions regarding family support. The measure used examines family support in a general sense, but one of the biggest problems for sexual minority youth is family support regarding being a sexual minority specifically. Someone could experience a tremendous amount of support at home for everything except being LGB, and not having that specific support could invalidate the buffer offered by the other kinds of support experienced at home. That said, it's still interesting that level of general support relates to experiences of SOM. Some of this might be because coming out can lead to reduced overall support, but that assumption is not a guarantee.

An important limitation to address is how gender was captured. Because this study targeted people who are sexual minorities, it's quite possible the authors also found people whose gender identities fall outside the binary of man and woman. If participants were only given two options to identify gender, people who are transgender, gender nonbinary, or genderqueer likely selected a gender identity that most closely reflected their identity (or the identity that they were initially assigned), but their gender identity was erased in the process. Yes, gender identity should be studied separately from sexual identity, but research on LGB communities will likely also include people who are not on the gender binary.

I have a couple minor editing points as well:

Page 2, line 57: change "warranted" to "warranting"

Line 67: in mixed what? Or the results were mixed?

Procedures: over what time period was this data collected?

Measures: make a statement that the SOMI structure generally follows the types of microaggressions (microassaults, microinvalidations, microinsults) because I know that three of the four factors do map onto that list (one was found to be unique to LGBQ populations)

Demographics: on page 3, line 123, you use the term "homosexual," and then on line 124 you use the term "a gay", how does gay/lesbian translate into Chinese, particularly given how "homosexual" versus "gay/lesbian" is used in English? Typically, homosexual is considered an outdated clinical term from the time when homosexuality was considered a mental disorder, whereas gay or lesbian are preferred terms to describe identity today. One also would not see the word gay used as a noun (e.g., "a gay) but rather as an adjective: "When did you first identify yourself as gay or bisexual?"

Because you provided assessments of mental health and suicidality, did you also provide participants with referrals to mental health resources if needed? (discuss research ethics)

Page 5, line 208-9: I thought in your literature review you indicated that findings were mixed for gender, not that they hadn't been studied before, and you also determined from your literature review that men would experience more SOM than women. Here you state that gender differences were not examined. Which is the case? I do see that the studies you cite refer to bullying and not SOM; it's worth being a bit more precise in your discussion as well. Bullying and microaggressions clearly overlap with each other, despite being different constructs. What you seem to add to this literature is that men experience more subtle homophobia than women in addition to more bullying.

Page 6, line 224: define polyvictimization

Line 224-7: can you unpack this statement about the increased risk of encountering SOM outside family environments when experiencing less family support? It obviously seems impossible for family support to cause more SOM experiences outside the family, so what factors would be correlated with family support, or mediated by family support, that would result in this increase? Is it possible that the increase in SOM by people in unsupportive family environments is simply additive of SOM experienced outside the family with those experienced inside the family? Or maybe someone is a bit more aware of all SOM when they don't experience family support (family support offers a buffer to minimize experiences of SOM outside the family)?

Page 6, line 232: missing "of" in "the enactors SOM might"

Thank you for this study!

Author Response

We appreciated your valuable comments. As discussed below, we have revised our manuscript with underlines based on your suggestions. Please let us know if we need to provide anything else regarding this revision.

Comment 1

My biggest concern is in the measurement of family support. I want to ask the authors to exercise caution in how they draw conclusions regarding family support. The measure used examines family support in a general sense, but one of the biggest problems for sexual minority youth is family support regarding being a sexual minority specifically. Someone could experience a tremendous amount of support at home for everything except being LGB, and not having that specific support could invalidate the buffer offered by the other kinds of support experienced at home. That said, it's still interesting that level of general support relates to experiences of SOM. Some of this might be because coming out can lead to reduced overall support, but that assumption is not a guarantee.

Response

We agree that general family support and family acceptance and rejection of sexual orientation may have various associations with SOM in LGB individuals. We explained what we measured was “general family support” thorough the revised manuscript. We also added it as one of limitations into Discussion as below. Please refer to line 286-291.

“Fifth, the present study examined the association of SOM with perceived general family support but not family acceptance and rejection of sexual orientation. Research found that family acceptance and rejection of sexual orientation is crucial to the health and well-being of LGB individuals, especially LGB youth [49]. The role of family acceptance and rejection of sexual orientation in the experience of SOM among LGB individuals warrants further study.”

Comment 2

An important limitation to address is how gender was captured. Because this study targeted people who are sexual minorities, it's quite possible the authors also found people whose gender identities fall outside the binary of man and woman. If participants were only given two options to identify gender, people who are transgender, gender nonbinary, or genderqueer likely selected a gender identity that most closely reflected their identity (or the identity that they were initially assigned), but their gender identity was erased in the process. Yes, gender identity should be studied separately from sexual identity, but research on LGB communities will likely also include people who are not on the gender binary.

Response

We agree that sexual minority and gender minority may have intersectional effects on SOM. We listed the lack of determining gender minority as one of the limitations as below. Please refer to line 283-286.

Fourth, this study inquired participants’ gender identities by the binary of man and woman but did not include the options of transgender, gender nonbinary, or genderqueer. Research has found that sexual and gender minority identities have intersectional impacts on health [49], behaviors [50], and risk of intimate partner violence [51].”

Comment 3

Page 2, line 57: change "warranted" to "warranting"

Response

We changed "warranted" to "warranting". Please refer to line 61.

Comment 4

Line 67: in mixed what? Or the results were mixed?

Response

Thank you for your reminding. We revised it into “the results of previous studies…were mixed.” Please refer to line 70.

Comment 5

Procedures: over what time period was this data collected?

Response

Thank you for your reminding. We missed the time period for microaggression and general family support. We added them as below into the revised manuscript.

  1. Microaggression: “in the last 6 months”. Please refer to line 116.
  2. General family support: “in recent one month”. Please refer to line 136.

Comment 6

Measures: make a statement that the SOMI structure generally follows the types of microaggressions (microassaults, microinvalidations, microinsults) because I know that three of the four factors do map onto that list (one was found to be unique to LGBQ populations)

Response

We added this statement as below into the revised manuscript. Please refer to line 118-119.

“The SOMI structure generally follows the types of microaggressions (microassaults, microinvalidations, microinsults) [4].”

Comment 7

Demographics: on page 3, line 123, you use the term "homosexual," and then on line 124 you use the term "a gay", how does gay/lesbian translate into Chinese, particularly given how "homosexual" versus "gay/lesbian" is used in English? Typically, homosexual is considered an outdated clinical term from the time when homosexuality was considered a mental disorder, whereas gay or lesbian are preferred terms to describe identity today. One also would not see the word gay used as a noun (e.g., "a gay) but rather as an adjective: "When did you first identify yourself as gay or bisexual?"

Response

Thank you for your reminding. We replaced “homosexual” by “gay/lesbian” in the revised manuscript. Please refer to lines96, 131 and 132 and Table 1-2.

Comment 8

Because you provided assessments of mental health and suicidality, did you also provide participants with referrals to mental health resources if needed? (discuss research ethics)

Response

Yes, we did provide the information of mental health resources to the participants. We added the statement as below into the revised manuscript. Please refer to line 109-111.

Because the present study assessed participants’ mental health and suicidality, we provided participants with the information of mental health resources if needed.”

Comment 9

Page 5, line 208-9: I thought in your literature review you indicated that findings were mixed for gender, not that they hadn't been studied before, and you also determined from your literature review that men would experience more SOM than women. Here you state that gender differences were not examined. Which is the case? I do see that the studies you cite refer to bullying and not SOM; it's worth being a bit more precise in your discussion as well. Bullying and microaggressions clearly overlap with each other, despite being different constructs. What you seem to add to this literature is that men experience more subtle homophobia than women in addition to more bullying.

Response

Thank you for your reminding. We revised the description aas below in the revised manuscript. Please refer to line 218-223.

“A previous qualitative study found that compared with lesbians, bisexual-identified women reported several bisexual-specific SOM, including hostility; denial/dismissal; unintelligibility; pressure to change; lesbian, gay, bisexual and transgender legitimacy; dating exclusion; and hypersexuality [40]. Further studies are needed to deeply explore the gender differences in the specific contexts of SOM experienced by LGB individuals.

Comment 10

Page 6, line 224: define polyvictimization

Response

“Polyvictimization” included “property victimization, bias and non-bias-motivated forms of physical assault, child maltreatment, sexual victimization, intimate partner violence, school-based bullying, cyberbullying, and indirect or witnessed forms of victimization.” We added into the revised manuscript. Please refer to line 236-239.

Comment 11

Line 224-7: can you unpack this statement about the increased risk of encountering SOM outside family environments when experiencing less family support? It obviously seems impossible for family support to cause more SOM experiences outside the family, so what factors would be correlated with family support, or mediated by family support, that would result in this increase? Is it possible that the increase in SOM by people in unsupportive family environments is simply additive of SOM experienced outside the family with those experienced inside the family? Or maybe someone is a bit more aware of all SOM when they don't experience family support (family support offers a buffer to minimize experiences of SOM outside the family)?

Response

Thank you for your suggestion. We added more discussion regarding the association between low family support and the experiences of SOM as below. Please refer to line 242-249.

“Young LGB individuals may need families’ assistance to meet developmental demands and to guide their personal experiences in various domains (e.g., interpersonal, romantic) and settings (e.g., school, work). LGB individuals with inadequate support from families may spend much time to be with and seek assistance from the individuals outside family environments; however, the chances of experiencing SOM may also increase simultaneously. Alternatively, LGB individuals experiencing greater SOM may need more family support for coping with such challenges; the chances to report perceived low family support may also increase.”

Comment 12

Page 6, line 232: missing "of" in "the enactors SOM might"

Response

Thank you for your reminding. We added “of” into the revised manuscript. Please refer to line 254.